# Translational efficiency across healthy and tumor tissues is proliferation-related

Xavier Hernandez-Alias[1] (ID), Hannah Benisty[1] (ID), Martin H Schaefer[1,2,*] (ID) & Luis Serrano[1,3,4,**] (ID)

## Abstract

Different tissues express genes with particular codon usage and anticodon tRNA repertoires. However, the codon–anticodon co-adaptation in humans is not completely understood, nor is its effect on tissue-specific protein levels. Here, we first validated the accuracy of small RNA-seq for tRNA quantification across five human cell lines. We then analyzed the tRNA abundance of more than 8,000 tumor samples from TCGA, together with their paired mRNA-seq and proteomics data, to determine the Supply-to-Demand Adaptation. We thereby elucidate that the dynamic adaptation of the tRNA pool is largely related to the proliferative state across tissues. The distribution of such tRNA pools over the whole cellular translatome affects the subsequent translational efficiency, which functionally determines a condition-specific expression program both in healthy and tumor states. Furthermore, the aberrant translational efficiency of some codons in cancer, exemplified by ProCCA and GlyGGT, is associated with poor patient survival. The regulation of these tRNA profiles is partly explained by the tRNA gene copy numbers and their promoter DNA methylation.

**Keywords** codon usage; The Cancer Genome Atlas; tissue; translation; tRNA
**Subject Categories** Computational Biology; Translation & Protein Quality
**Mol Syst Biol. (2020) 16: e9275**

## Introduction

In the light of the genetic code, multiple 3-letter combinations of nucleotides in the mRNA can give rise to the same amino acid, which are known as synonymous codons. However, despite the homology at the protein level, these different codons are recognized distinctly by the transcriptional and translational machineries (Supek, 2016; Hanson & Coller, 2017) and ultimately cause changes at multiple levels of gene expression. Therefore, the non-uniform abundance of synonymous codons across different tissues and among distinct functional gene sets has been proposed as an adaptive mechanism of gene expression regulation (Najafabadi et al, 2009), particularly linked to the proliferative state (Gingold et al, 2014). Nevertheless, in human, it is still under debate whether the efficiency of gene expression is the main selective pressure driving the evolution of genomic codon usage (Pouyet et al, 2017).

The 61 amino-acid-coding codons need to be recognized by 46 different tRNA isoacceptors distributed across 428 Pol-III-transcribed tRNA genes (Chan & Lowe, 2016), thus requiring wobble interactions (non-Watson-Crick base pairing). This complexity of the tRNA repertoire is further enhanced by an average of 11–13 base modifications per tRNA and all possible combinations thereof (Schimmel, 2018). The underlying mechanisms regulating tRNA gene expression and modification are far from resolved (Pan, 2018; Rak et al, 2018). However, it has been established that different conditions and tissues showcase distinct tRNA abundances (Dittmar et al, 2006; Gingold et al, 2014) and codon usages (Najafabadi et al, 2009; Waldman et al, 2010).

In order to understand such changes in codon–anticodon co-adaptation, orthogonal datasets of gene expression including tRNA quantification are required, which needs to overcome the challenges of strong secondary structures and abundant chemical modifications. Recent technological developments have paved the way for sensitive high-throughput tRNA sequencing across tissues and conditions (Zheng et al, 2015a; Gogakos et al, 2017a). Aside from these methods and despite the lower coverage, tRNA reads can also be detected from generic small RNA-seq datasets (Guo et al, 2015, 2016; Pundhir & Gorodkin, 2015; Torres et al, 2015a; Hoffmann et al, 2018). In this context, The Cancer Genome Atlas (TCGA) has been recently used to investigate the alteration of tRNA gene expression and translational machinery in cancer, which may play a role in driving aberrant translation (Zhang et al, 2018, 2019).

To validate the use of small RNA-seq for tRNA quantification, we first compare tRNA levels determined in HEK293 by well-established tRNA sequencing methods (Hydro-tRNAseq and demethylase-tRNAseq) (Zheng et al, 2015a; Gogakos et al, 2017a; Mattijssen et al, 2017a), with those obtained by small RNA-seq. Then, we quantify the tRNA repertoire of five cell lines using Hydro-tRNAseq and perform small RNA-seq in parallel. Comparison of the tRNA

1 Centre for Genomic Regulation (CRG), The Barcelona Institute of Science and Technology, Barcelona, Spain
2 Department of Experimental Oncology, IEO, European Institute of Oncology IRCCS, Milan, Italy
3 Universitat Pompeu Fabra (UPF), Barcelona, Spain
4 ICREA, Barcelona, Spain
*Corresponding author. Tel: +39 0294372331; E-mail: martin.schaefer@ieo.it
**Corresponding author. Tel: +34 933160101; E-mail: luis.serrano@crg.eu

measures obtained by both approaches shows that it is possible to accurately estimate relative tRNA abundance of cells and tissues using small RNA-seq. Furthermore, we show that both types of quantification are informative enough to distinguish between the five analyzed human cell lines covering multiple tissue types. In consequence, we apply a tRNA-specific computational pipeline to re-analyze 8,534 small RNA-seq datasets from TCGA (Chu *et al*, 2016). We find that the tissue specificity of tRNA profiles is largely proliferation-related, even within healthy tissues. The tRNA quantification of TCGA samples enables their comparison with paired and publicly available mRNA-seq, proteomic, DNA methylation, and copy number data, which underscores the role of tRNAs in globally controlling a condition-specific translational program. We discover multiple codons, including ProCCA and GlyGGT, whose translational efficiency is compromised and leads to poor prognosis in cancer. Finally, promoter DNA methylation and tRNA gene copy number arise as two regulatory mechanisms controlling tRNA abundances in cancer.

# Results

## tRNA quantification and modifications from small RNA-seq data

In order to test how accurately we can extract tRNA abundance information contained in small RNA sequencing data, we re-analyze four publicly available datasets of the cell line HEK293 (Flores *et al*, 2014a; Data ref: Flores *et al*, 2014b; Mefferd *et al*, 2015a; Data ref: Mefferd *et al*, 2015b; Torres *et al*, 2015a; Data ref: Torres *et al*, 2015b,c). In contrast to previous studies analyzing tRNA expression from small RNA-seq data (Zhang *et al*, 2018, 2019), we use a computational pipeline specifically developed for the accurate mapping of tRNA reads (Hoffmann *et al*, 2018) in order to quantify all different isoacceptor species (Fig 1A, see Materials and Methods). To validate the accuracy of these small RNA-seq quantifications, we retrieve four datasets of well-established tRNA sequencing methods (Hydro-tRNAseq and demethylase-tRNAseq) applied to the same cell type (Zheng *et al*, 2015a; Data ref: Zheng *et al*, 2015b; Gogakos *et al*, 2017a; Data ref: Gogakos *et al*, 2017b; Mattijssen *et al*, 2017a; Data ref: Mattijssen *et al*, 2017b; preprint: Benisty *et al*, 2019a; Data ref: Benisty *et al*, 2019b), which autocorrelate in the range of 0.75–0.85 among themselves (Table EV1, Fig EV1A). In comparison, our four HEK293 small RNA-seq quantifications show an average Spearman correlation against these four conventional datasets of 0.73. Compared to the Zhang *et al* (2018) quantification, which correlate in the range of 0.60–0.77 (Table EV1, Fig EV1A), our tRNA-specific mapping pipeline performs slightly better than the previously published protocol. It has been reported that there are tRNA-derived fragments naturally produced and having other functions different from translation (Schimmel, 2018), which could confound the tRNA quantification. Although we cannot exclude the presence of tRNA-derived fragments in small RNA-seq datasets (Torres *et al*, 2019), we found that no differences between reads with or without mismatches are found when compared to tRNAseq protocols in which tRFs are specifically removed before sequencing.

Further than correlating small RNA-seq data with conventional tRNAseq datasets, we analyze whether small RNA-seq quantifications are informative enough to distinguish between different human cell lines covering multiple tissue types. We therefore apply both small RNA-seq and Hydro-tRNAseq to HEK293 (kidney), HCT116 (colon), HeLa (cervix), MDA-MB-231 (breast), and BJ fibroblasts. However, given the high variability between replicates of MDA-MB-231 Hydro-tRNAseq quantifications, this cell line was excluded from further analyses (Table EV2). First, the correlations between the two methods of identical samples and computational mapping pipeline range between 0.93 and 0.96 for all cell lines. tRNA quantifications from both protocols are compared, and significantly higher Spearman correlations are obtained within matching samples versus mismatching cell lines (Fig 1B). In order to assess the amount of tRNA variability coming from either the sequencing method or the cell lines, a principal component analysis of these tRNA quantifications indicates that both factors influence variability to a similar extent (Fig EV1B, > 30% variance each). Furthermore, to validate that small RNA-seq is similarly informative of cell type differences as Hydro-tRNAseq, we show that a linear discriminant analysis of the data is able to discriminate between cell lines regardless of the sequencing protocol (Fig EV1B).

We also detect tRNA base modifications in both protocols by nucleotide variant calling, as described in Hoffmann *et al* (2018). In all cases, considering the modifications that are detected in all three replicates, Hydro-tRNAseq datasets identify a larger number of modifications than small RNA-seq, as expected by the more uniform and deeper coverage of this method (Fig EV2, Table EV2). Furthermore, we detect a significant enrichment of the Hydro-tRNAseq modifications in the small RNA-seq data (*P* < 1e-16, Fisher's test), indicating that the latter contains also information on tRNA modifications (Fig 1C). Although the exact nature of modifications cannot be determined by sequencing, most frequent nucleotide mismatches in both sequencing methods include A-to-G changes at position 34 and 37 (Fig EV3, Table EV3), which correspond to known modifications such as adenosine-to-inosine editing and 1-methylinosine, respectively (Pan, 2018). Overall, most of the known modification-specific mismatches can be retrieved with both small RNA-seq and Hydro-tRNAseq (Table EV3), while the deeper coverage of the latter improves its sensitivity.

Taken together, these observations demonstrate the applicability of small RNA-seq data for the quantification of tRNAs and their modifications. We therefore apply the same computational pipeline to all healthy and primary tumor small RNA-seq samples from 23 cancer types of The Cancer Genome Atlas (TCGA), which consists of 8,605 samples distributed among 17 different human tissues (Fig 1D, number of samples and their abbreviations in Table EV4).

## Proliferation is the major driver of tissue specificity in tRNAs

To determine the tissue specificity of tRNAs in physiological conditions, the tRNA levels of all 675 healthy samples in TCGA tissues are analyzed from small RNA-seq data. For all 46 annotated anticodons, tRNA abundances have significant differences between tissues, as detected by Kruskal–Wallis test (*q* < 0.05, FDR-corrected). Such differences between tissues are also observed by hierarchical clustering of the median abundance between all groups (Fig 2A). Furthermore, healthy samples from cancer types originating from the same tissue tend to cluster together: READ and COAD from the gut; KIRC, KIRP, and KICH from the kidney; LUAD and

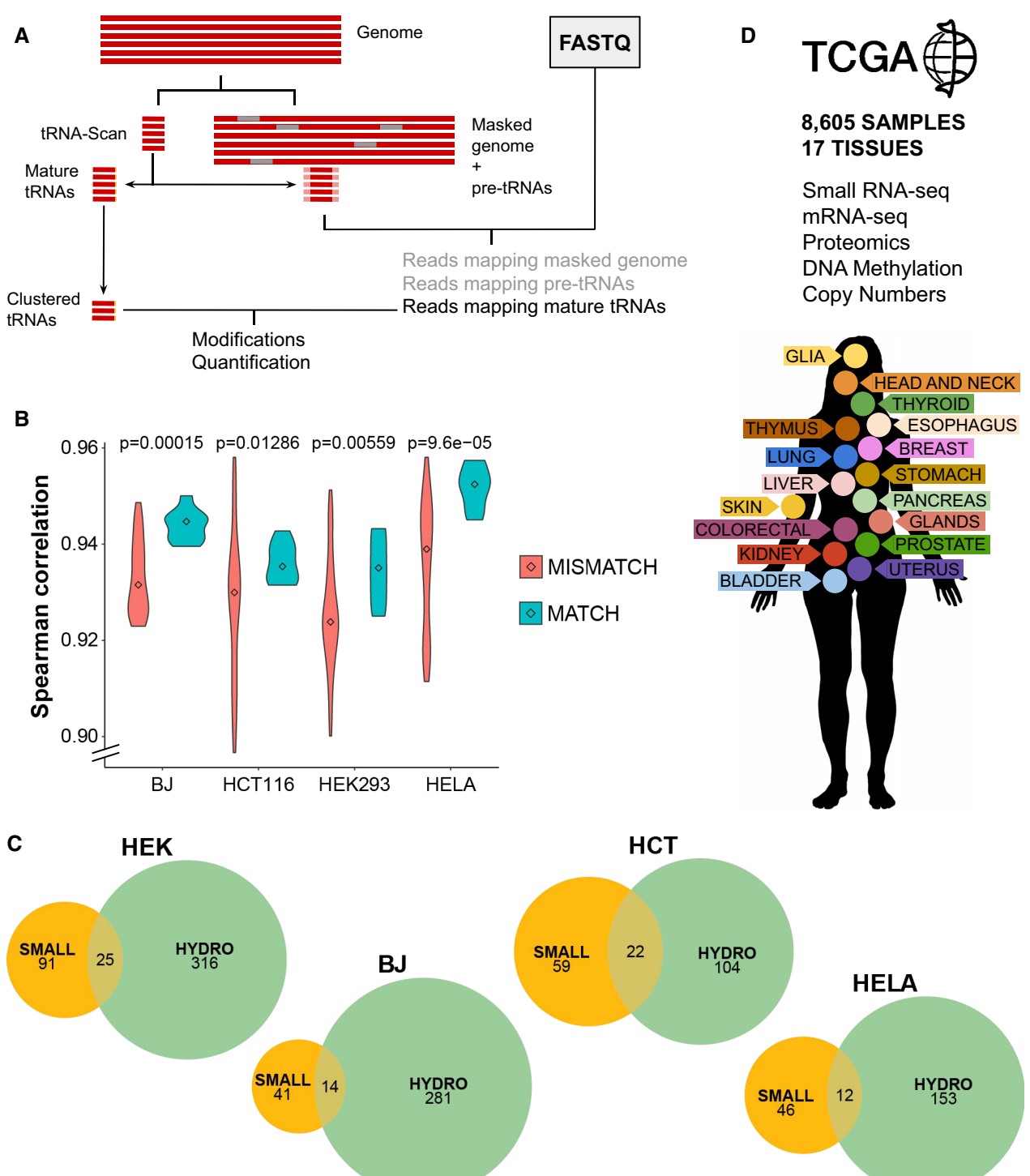

**Figure 1. tRNA quantification and modifications from small RNA-seq data.**

A  Schematic pipeline for accurate mapping of tRNA reads.
B  Correlations between tRNA quantifications by small RNA-seq and Hydro-tRNAseq of matching (correlations within the same cell line) versus non-matching (different cell lines) samples. Center values represent the median. The *P*-value corresponds to a one-tailed Wilcoxon rank-sum test, with $n_{matching} = 9$ and $n_{non-matching} = 63$.
C  Overlap of the detected tRNA modifications upon variant calling by both methods.
D  The TCGA network contains small RNA-seq data alongside mRNA-seq, DNA methylation arrays, non-targeted proteomics, and copy number alteration quantification comprising 17 tissues.

Source data are available online for this figure.

LUSC from the lung; UCEC and CESC from the uterus; and LIHC and CHOL from the liver (refer to Table EV4 for full cancer names). On the other hand, in terms of anticodon abundances, three main subgroups of tRNAs with low, medium, and high levels can be distinguished across all cancer types (Fig 2A).

Regarding codon usage, a measure of tRNA abundance taking into account the relative contribution of each tRNA anticodon among the set of synonymous codons of a certain amino acid is the relative anticodon abundance (see Materials and Methods). Using this measure, a principal component analysis (PCA) of all healthy control samples in TCGA also shows clear differences between tissues (Fig 2B). To interrogate the biological functions related to the variability of anticodon abundances between samples, we compute the correlation of the whole mRNA-seq transcriptome against the first PCA component, which explains 18.5% of the variance, and analyze it by gene set enrichment analysis (GSEA). As a result, the top correlating genes are enriched in proliferation and immune cell activation, while the lowest correlations belong to genes related with oxidative metabolism and respiration (Fig 2C, Table EV5). Moreover, our first component correlates positively with the proliferation marker Ki67 ($R_{spearman} = 0.45$) (Scholzen & Gerdes, 2000). This confirms, as has been previously suggested (Gingold *et al*, 2014), that there is a proliferative tRNA expression program.

Overall, we observe patterns of tissue-specific tRNA profiles in TCGA healthy samples. Furthermore, based on both the gene set enrichment and the association to a proliferation marker, our analyses identify the proliferative state of tissues as the major biological function driving the variability on tRNA abundances.

## tRNA repertoires determine tissue-specific translational efficiency

Given that different tissues express distinct tRNA repertoires, we wondered whether they could have an effect in protein translation elongation. The so-called translational efficiency is defined as the rate of protein production from mRNA, and multiple indices and models can be described to estimate it (Gingold & Pilpel, 2011). In this article, and based on previous studies underscoring the global control role of codon usage as a competition for a limited tRNA pool (Gingold *et al*, 2012; Pechmann & Frydman, 2013; Frumkin *et al*, 2018), we define the Supply-to-Demand Adaptation (SDA) as the balance between the supply (i.e., the anticodon tRNA abundances) and demand (i.e., the weighted codon usage based on the mRNA levels) for each of the 60 codons (excluding methionine and stop codons). Furthermore, we normalize both the codon and anticodon abundances within each amino acid family (i.e., relative to the most abundant synonymous codon/anticodon), in order to remove the effect of amino acid biases and get a cleaner measure of codon optimality (Eraslan *et al*, 2019).

To validate the suitability of SDA in determining the translational efficiency, we correlate the SDA value of all proteins against the available proteomics data of paired TCGA samples (Slebos *et al*, 2015; Mertins *et al*, 2016), which includes breast and colorectal tissues (tumor only, as no healthy samples are available). Although correlations are modest, both the protein abundances and the protein-to-mRNA ratios correlate significantly better with SDA than with the classical tRNA Adaptation Index [tAI] (dos Reis *et al*, 2003,

2004) or with a relative tAI with normalized weights within each amino acid family [RtAI] (Figs 3A and EV4A and B). In consequence, including the mRNA codon demand into the SDA metric outperforms other tRNA-only metrics of translational efficiency. Furthermore, the correlation of SDA with protein-to-mRNA ratio is slightly but significantly higher than with protein levels alone, which indicates that the first is a better proxy for the process of translation (Fig 3A).

Next, we calculate the SDA for the 620 healthy samples for which both tRNA abundances and mRNA levels are available. When analyzing the tissue medians of SDA weights per each codon (SDAw), we observe that most codons are optimally balanced (SDAw = 1), while 12.4 and 23.6% of codons are favored (SDAw > 2) and disfavored (SDAw < 0.5), respectively. The tissue clustering again shows that healthy samples of cancer types from the same tissue have similar SDAw profiles, which separates two major clusters of mostly high-Ki67 and low-Ki67 tissues (Fig EV4C).

In order to identify the codons contributing most to the differences between tissues, we compute a bidimensional PCA across all samples and SDAw (Fig 3B). Both the first and second components significantly correlate with the proliferation marker Ki67 (0.4 and 0.35; see Fig 3B). In agreement with the proliferation- and differentiation-related codons of Gingold *et al* (2014), such proliferative pattern is similarly reproduced by the codons contributing to the first PCA component, which has the strongest association to proliferation (Fig 3B). Further, similarly to the tRNA abundances (Fig 2B), a GSEA of correlating genes with the first component highlights the link with proliferation-related terms (Table EV6). On the other hand, the first component also clearly separates codons based on the GC content of the third codon base, which has recently been associated with differentiation (high in nnC/G codons) versus self-renewal functions (high in nnA/T) (Bornelöv *et al*, 2019), as well as with proliferative transcriptomes (Fornasiero & Rizzoli, 2019).

The previous analyses support the idea of proliferation-related tRNAs driving changes in translational efficiencies. In that case, we expect that the two most extreme tissues in terms of proliferation (brain and gut, excluding thymus for its low number of samples) differ in the optimization of proliferation-related proteins. As such, we compute the average SDAw for these two tissues, analyze the subsequent SDA score for each protein, and perform a GSEA of the differential SDA per protein. Consistent with our hypothesis, the results indicate that gut-optimized proteins are enriched in translation, DNA replication, and protein localization, whereas brain-optimized proteins are related to phospholipid production and neural function (Fig 3C, Table EV7). Taken together, this result confirms that the tRNA-dependent translational efficiency is optimized for the translation of tissue-specific genes, particularly in function of the proliferation state.

## Aberrant translational efficiencies drive tumor progression

Given that proliferation is a major determinant of translational efficiency in healthy tissues, its importance could be extrapolated to pathological conditions such as cancer. In fact, aberrant expression of tRNAs and codon usage have been broadly related with tumorigenesis and cancer progression (preprint: Benisty *et al*, 2019a; Goodarzi *et al*, 2016; Zhang *et al*, 2018, 2019). We therefore

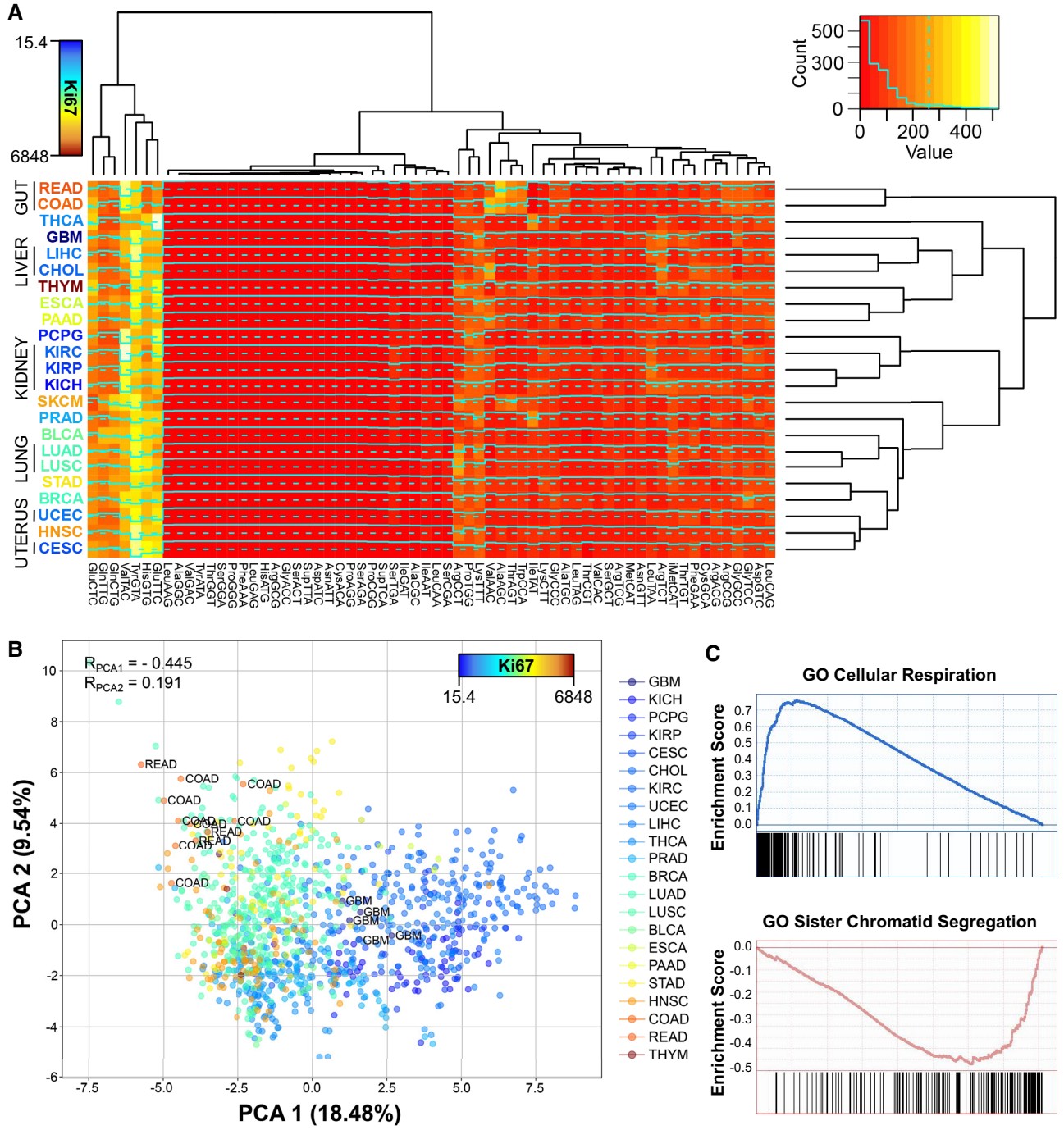

**Figure 2. Proliferation is the major driver of tissue specificity in tRNAs.**

A   Medians of square-root-normalized tRNA abundances across all TCGA tissues. The color of the tissue labels corresponds to the average Ki67 expression. Refer to Table EV4 for full cancer type names and number of samples.

B   Principal component analysis (PCA) of the relative anticodon abundances (RAA, see Materials and Methods) of all healthy samples of TCGA, where the color scale corresponds to the mean tissue expression of Ki67. The Spearman correlations of Ki67 with the components are shown, as well as the samples of most extreme tissues.

C   Top positive and negative GO terms upon gene set enrichment analysis (GSEA) of the correlations of the first PCA component against all genes.

Data information: Abbreviations stand for BLCA (bladder urothelial carcinoma), BRCA (breast invasive carcinoma), CESC (cervical squamous cell carcinoma and endocervical adenocarcinoma), CHOL (cholangiocarcinoma), COAD (colon adenocarcinoma), ESCA (esophageal carcinoma), GBM (glioblastoma multiforme), HNSC (head and neck squamous cell carcinoma), KICH (kidney chromophobe), KIRC (kidney renal clear cell carcinoma), KIRP (kidney renal papillary cell carcinoma), LIHC (liver hepatocellular carcinoma), LUAD (lung adenocarcinoma), LUSC (lung squamous cell carcinoma), PAAD (pancreatic adenocarcinoma), PCPG (pheochromocytoma and paraganglioma), PRAD (prostate adenocarcinoma), READ (rectum adenocarcinoma), SKCM (skin cutaneous melanoma), STAD (stomach adenocarcinoma), THCA (thyroid carcinoma), THYM (thymoma), UCEC (uterine corpus endometrial carcinoma).

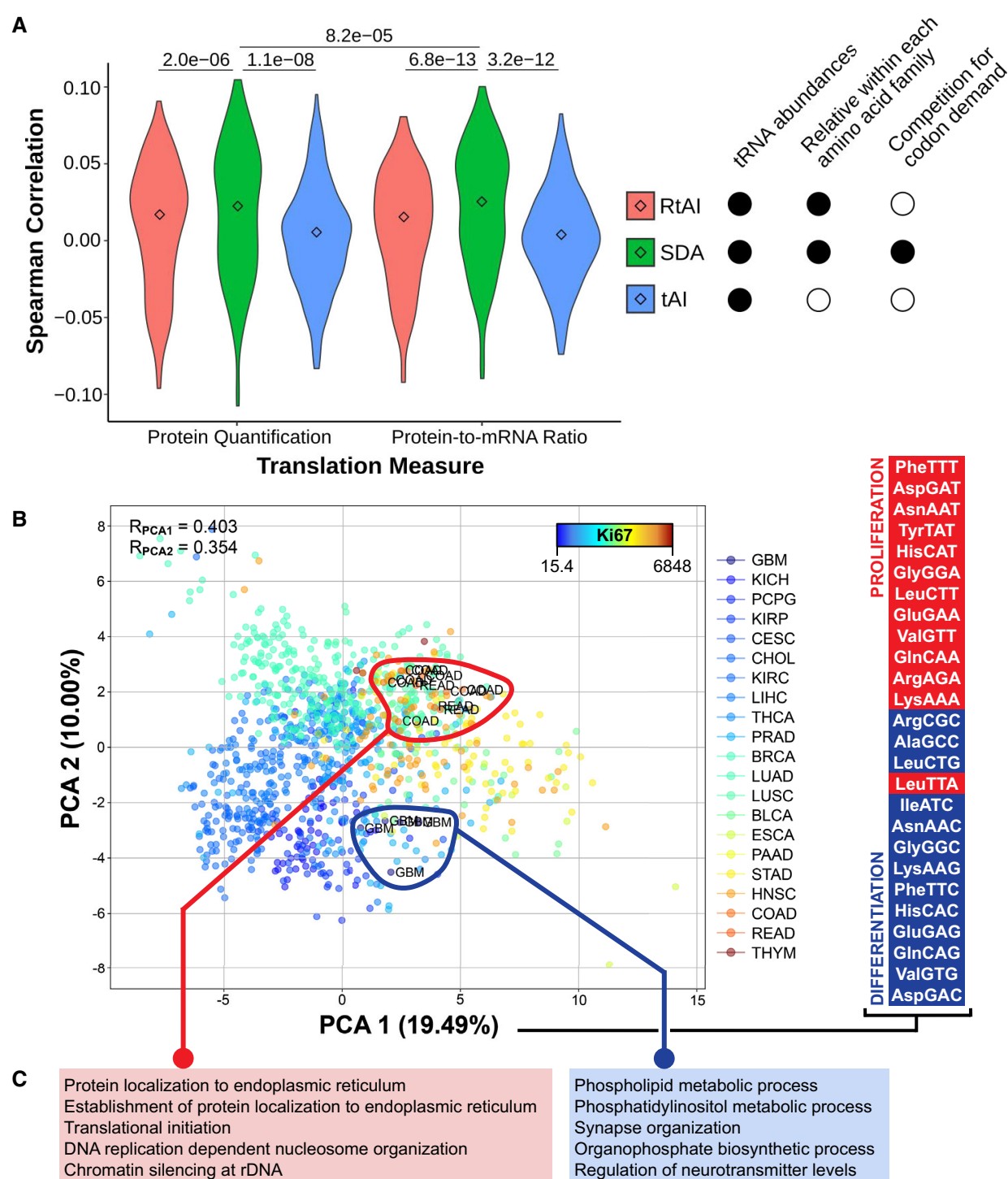

**Figure 3. tRNA repertoires determine tissue-specific translational efficiency.**

A   Three metrics of translational efficiency (the classical tAI, a relative tAI with normalized weights within each amino acid family, and the Supply-to-Demand Adaptation described in this article) are Spearman correlated against two proxies of translation (protein abundance and protein-to-mRNA ratio) for all samples for which proteomics data are available (BRCA, COAD and READ). Center values represent the median. Statistical differences are determined by sample-paired two-tailed Wilcoxon rank-sum test ($n = 219$).

B   Principal component analysis (PCA) of the SDAw of TCGA, where the color scale corresponds to the mean tissue expression of Ki67. The Spearman correlations of Ki67 with the components are shown, as well as the samples of most extreme tissues. On the right, the top and bottom proliferation- and differentiation-related codons, as defined by Gingold *et al* (2014), ordered by their contribution to the first PCA component. Refer to Table EV4 for full cancer type names and number of samples.

C   GSEA of the differential SDA between extreme tissues ($\Delta SDA = SDA_{Colorectal} - SDA_{Brain}$), showing the top five GO terms with high (left) and low (right) SDA in colorectal versus glial tissues.

investigate 22 cancer types from TCGA in order to determine which codons are translationally compromised in disease.

Similar to the analysis performed on the healthy tissues, we quantify all tRNA abundances of TCGA primary tumor samples (Fig EV5) and determine their corresponding translational efficiencies using the SDA metric. By analyzing the differential SDAw between normal and tumor samples, we observe many significant differences in all 60 codons across the 22 cancer types (Fig 4A). Among the most consistent changes, the ProCCA codon is significantly more favored in tumors for 8 out of 10 cancer types, while the ProCCG is disfavored in 14 out of 16 cancers (Fig 4B). In the case of glycine, GlyGGT is better adapted in healthy samples (13/13), whereas tumor mostly favors GlyGGC (9/12) and GlyGGG (7/9).

In terms of patient survival, we divide the TCGA patients in two groups based on their low or high tumor SDAw and analyze their survival probability (Fig 4C, Table EV8). Among others, and consistent with the previous analysis, high supply-to-demand weights of ProCCA are associated with poor prognosis in kidney renal clear cell carcinoma and kidney renal papillary cell carcinoma. Proline limitation in clear cell renal cell carcinoma has been shown to compromise CCA-decoding tRNAPro aminoacylation, leading to reduced tumor growth (Loayza-Puch et al, 2016). In contrast, high SDAw of GlyGGT and ValGTC lead to longer survival in kidney chromophobe and head and neck squamous cell carcinoma, respectively.

To determine the impact of aberrant translational efficiencies in regulating an oncogenic translation program, we calculate the differential SDA for the whole genome based on the average SDAw of healthy and tumor samples in kidney renal clear cell carcinoma, since it is the cancer type with the most SDAw differences (Fig 4A). The GSEA of the resulting ΔSDA score indicates that cancer SDAw should favor the translation of proteins related to DNA replication and gene expression, whereas the healthy kidney samples favor development and differentiation processes (Table EV9). As the SDAw of the ProCCA is specifically disturbed in cancer, we also interrogate how this codon is distributed along the genome. We therefore perform a GSEA on the relative codon usage of ProCCA, which shows that DNA replication and cell cycle functions lie among the most CCA-enriched genes, while morphogenesis and differentiation terms are CCA-depleted (Table EV10). Together with the low-proliferative state of kidney (Fig 2B), the over-efficiency of a proliferation-related codon in this tissue can thus perturb its cellular SDA.

Overall, we detect differences at the level of SDAw between tumor and healthy tissues, which show a functional relevance to the disease state. Therefore, while the differential expression of tRNAs in TCGA had been already discussed elsewhere (Zhang et al, 2018, 2019), we could here elucidate their oncogenic effect in translational efficiency. In particular, ProCCA appears as an interesting codon candidate in favoring tumor progression, which we had also detected in healthy tissues to be associated with proliferation (Fig 3B, Table EV6).

### Promoter methylation and gene copy number regulate the tRNA abundance

Aberrant translational efficiencies in cancer are partially caused by the differential abundance of tRNA genes (Fig EV5). To determine the underlying mechanisms driving changes in expression, we retrieve the DNA methylation (typically occurring at CpG dinucleotides) and copy number alteration (CNA) data from TCGA samples, as a possible means for tRNA gene regulation. While CNA information covers 84% of tRNA genes, the 450K BeadChip methylation arrays used in TCGA are mostly centered on the coding genome (Bibikova et al, 2011) and yield a coverage of only 37%.

In order to make the gene-based data comparable with the measured isoacceptor-based tRNA abundances, we average methylation and CNA levels over all genes within the same isoacceptor family, at the cost of losing resolution. For each isoacceptor and each cancer type, we finally fit a multiple linear regression to determine how are promoter methylation and CNA affecting tRNA expression (Fig 5A, Table EV11). Among all models, the significant coefficients for methylation and CNA are significantly negative and positive, respectively. Despite the limited explained variance of the models (average $R^2 = 0.023$), such results suggest that promoter methylation could contribute to inhibition of tRNA gene expression, whereas an increase in the gene copy number would enhance tRNA expression.

Given the association of the codon ProCCA with cancer prognosis (Fig 4C), we explore the abundance pattern of tRNAPro in TCGA. While both tRNAPro[TGG] and tRNAPro[AGG] are able to decode ProCCA, the latter specifically appears overexpressed in 8 out of 9 cancer types (Fig EV5A), making it a candidate driver of the translational differences. To get a more accurate picture of the tRNA gene methylation levels, we also analyze recently published bisulfite sequencing data (Zhou et al, 2018), which, for 47 samples among nine cancer types, improved the coverage of tRNA genes up to an average of 81%. In total, tRNAPro[AGG] genes stand among the most duplicated and least methylated proline isoacceptors in cancer (Fig EV6A and B), in particular at the chr6.tRNA12 and chr16.tRNA12 genes (Fig 5B). Furthermore, tRNAPro[AGG] gene duplications occur most frequently in kidney cancers (Fig EV6C). On the other hand, although the other CCA-decoding tRNAPro[TGG] is not differentially expressed in cancer (Fig EV5), its genes are as similarly methylated and duplicated as tRNAPro[AGG] (Figs 5B and EV6).

In short, promoter methylation and CNA appear as two possible regulatory mechanisms of tRNA expression in cancer, which suggests that similar mechanisms that control the Pol-II-mediated RNAs might also regulate the expression of Pol-III non-coding transcriptome, such as tRNA genes. However, more accurate and high-throughput data on the methylation and CNA of the non-coding genome together with gene-based tRNA quantifications are needed to make stronger associations.

## Discussion

In this study, we use a systems biology approach to interrogate the multi-omics TCGA dataset under the perspective of translational efficiencies. We therefore first validate the suitability of small RNA-seq data in reproducing conventional tRNAseq quantifications based on a gold standard set of five tissue-wide human cell lines. In fact, knowing that small RNA-seq datasets have a limited tRNA coverage and tend to be biased toward tRNA fragments and unmodified tRNAs (Torres et al, 2015a, 2019), we extend and apply a computational pipeline for accurate mapping of tRNA reads (Hoffmann et al, 2018). As a result, we obtain reproducible and informative quantifications of all isoacceptors in our gold standard cell lines as well as in thousands of samples across 23 cancer types of TCGA, exceeding the quality of

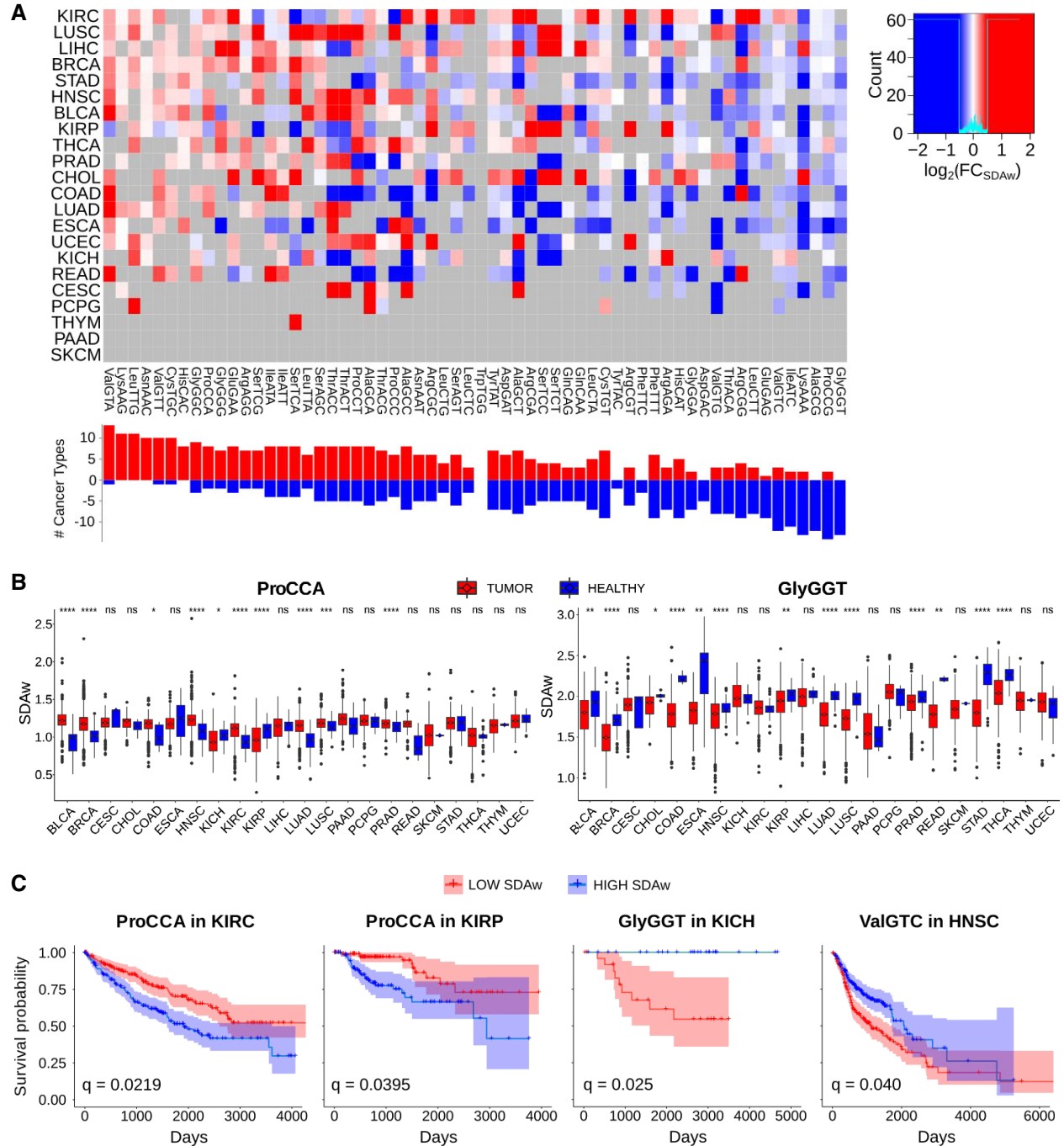

**Figure 4. Aberrant translational efficiencies drive tumor progression.**

A   Differential SDAw between healthy and tumor samples across 22 cancer types, as measured by $\log_2(\text{SDAw}_{\text{Tumor}}/\text{SDAw}_{\text{Healthy}})$. Only significant differences are colored, which are determined using a two-tailed Wilcoxon rank-sum test and corrected for multiple testing by FDR. Refer to Table EV4 for full cancer type names and number of samples.

B   Boxplot of the SDAw of ProCAA and AlaGCG codons across TCGA cancer types. Boxes expand from the first to the third quartile, with the center values indicating the median. The whiskers define a confidence interval of median $\pm$ 1.58*IQR/sqrt(n). Statistical differences are determined using a FDR-corrected two-tailed Wilcoxon rank-sum test ns ($P > 0.05$), * ($P \leq 0.05$), ** ($P \leq 0.01$), *** ($P \leq 0.001$), **** ($P \leq 0.0001$).

C   Survival curves for the previous codons in KIRC, KIRP, and BLCA patients. The survival analysis was performed for all codons whose SDAw were significantly different in more than 5 cancer types in the one direction with respect to the other [Abs(UP-DOWN) > 5] and correspondingly corrected for multiple comparisons using FDR.

similarly published data (Zhang *et al*, 2018, 2019). However, we cannot exclude that tRNA-derived fragments (tRFs) could be interfering with our small RNA-seq quantifications. At the level of nucleotide

modifications (Pan, 2018), our tRNA mapping pipeline is also able to detect most of the known mismatch-producing modifications of mature tRNAs. All in all, even though our quantifications from small

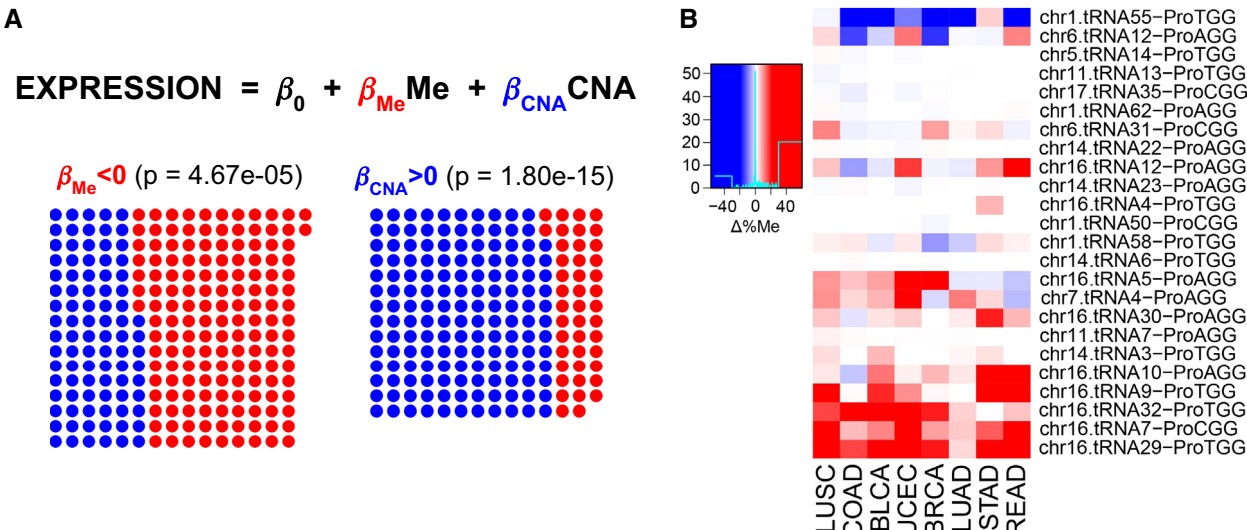

**Figure 5. Promoter methylation and gene copy number regulate tRNA abundance.**

A  A multiple linear regression (MLR) between square-root-normalized tRNA abundance and the average promoter methylation (450K BeadChip array) and gene copy number at the isoacceptor level. Among all MLRs for each isoacceptor and each cancer type separately, the dots show the FDR-normalized significant coefficients based on their corresponding $t$-statistic $P$-value, and red/blue shows whether they are negative/positive, respectively. The $P$-value corresponds to a two-tailed binomial test between $n_{pos}$ and $n_{neg}$.

B  Differential promoter methylation (bisulfite sequencing) between healthy and tumor samples of genes expressing proline tRNAs, as measured by $\Delta\%Me = (\%Me_{Tumor} - \%Me_{Healthy})$. Refer to Table EV4 for full cancer type names and number of samples.

RNA-seq just give an estimate of the tRNA abundances, the results indicate that they can be rather precise proxies.

From these quantifications, we then elucidate their effect on the translational efficiency by defining the SDA, for Supply-to-Demand Adaptation, which is a balance between the tRNA supply and the codon demand. Although a more accurate SDA would have determined the supply and demand based on the aminoacylated portion of tRNAs (Evans *et al*, 2017) and the ribosome-bound mRNAs (Ingolia *et al*, 2009), respectively, we approximate such measures by our tRNA quantifications and the publicly available mRNA-seq data of TCGA. In agreement with current studies showing that a dynamic codon usage needs to compete for a limited tRNA pool (Frumkin *et al*, 2018; Eraslan *et al*, 2019), we demonstrate that SDA is better measure of codon optimality than previously published metrics such as the tAI (dos Reis *et al*, 2003, 2004). However, far from explaining the translation process, the still low but significant correlations of protein-SDA in human, in contrast to unicellular organisms, suggest that protein expression is also dependent on other layers of regulation, such as transcriptional and post-transcriptional machineries, translation initiation, epigenetic modifications of DNA and RNAs, or protein degradation mechanisms (Rudolph *et al*, 2016).

On the level of translational efficiency, in agreement with previous studies (preprint: Benisty *et al*, 2019a; Gingold *et al*, 2014), we detect that the proliferative state is the major determinant of SDA differences both across healthy tissues and in cancer. Moreover, in contrast to recent work challenging the tissue specificity of codon–anticodon co-adaptation in human (Rudolph *et al*, 2016; Eraslan *et al*, 2019), our data here support the idea that tissue-specific SDAw have functional implications on the tissue phenotype (e.g., in favoring neural differentiation in brain or abnormal proliferation in

cancer). Furthermore, we observe a pattern of proliferative nnA/T versus differentiative nnC/G codons. Based on ribosome profiling experiments of pluripotency changes in embryonic stem cells (Bornelöv *et al*, 2019), this could be attributed to the slower translation in differentiated cells of codons decoded by tRNAs that require adenosine-to-inosine modification at the wobble-base pairing position. In particular, we detect the ProCCA codon to be significantly more favored in proliferative cells and leading to poor cancer prognosis in kidney carcinomas, specifically driven by an overexpression of tRNAPro$^{AGG}$ in cancer. Proline limitation in clear cell renal cell carcinoma has indeed been shown to mostly compromise tRNA-Pro$^{AGG}$ aminoacylation, leading to slower proline translation and reduced tumor growth (Loayza-Puch *et al*, 2016). Furthermore, in support of our approach for isoacceptor quantification and translational efficiency, similar studies of tRNA levels in TCGA have controversially claimed an opposite prognostic value for the ProCCA codon in clear renal cell carcinoma (Zhang *et al*, 2018, 2019).

In an effort to elucidate the mechanisms regulating the expression of tRNAs, we observe that the tRNA gene copy number and their DNA methylation state have a positive and inhibitory association with tRNA abundances, respectively. In this context, DNA methylation has previously been linked to the silencing of type II genes (such as tRNAs) of the Pol-III transcriptome (Besser *et al*, 1990; Park *et al*, 2017). Here, we specifically propose a role for DNA methylation in regulating the overexpression of tRNAPro$^{AGG}$ in cancer, although no direct causal link can yet be established. In terms of the copy number alterations, it is not surprising to detect tRNA gene duplications in tumors, but the functional role in disease of different isodecoder genes that share the same anticodon is still a matter of debate (Lant *et al*, 2019). With the advent of more accurate and high-throughput multi-omics datasets, our

knowledge on the underlying mechanisms controlling tRNA expression, degradation, and the effect of their modifications will be further expanded (Pan, 2018; Rak *et al*, 2018). Recent studies in TCGA have actually observed an upregulation of tRNA-modifying enzymes, as well as proposed a link of tRNA-derived fragments (tRF) to proliferation (Zhang *et al*, 2018; Telonis *et al*, 2019).

Overall, this is the first high-throughput study of codon–anticodon translational efficiency over thousands of samples comprising multiple tissues and disease. We therefore demonstrate a functional role for the proliferation-driven tRNA abundance differences in determining a tissue-specific phenotype, both in physiological and pathological conditions. In the future, we expect to validate the effect of such differential translational efficiency by integrating perturbation-based data and including additional gene expression regulatory layers such as tRNA modifications.

# Materials and Methods

## Reagents and Tools table

| Reagent/Resource | Reference or source | Identifier or catalog number |
|---|---|---|
| **Chemicals, enzymes, and other reagents** | | |
| Antarctic phosphatase | New England BioLabs | Cat#M0289 |
| T4 Polynucleotide Kinase | New England BioLabs | Cat#M0201 |
| ProtoScript II Reverse Transcriptase | New England BioLabs | Cat#M0368 |
| miRNeasy Mini kit | Qiagen | Cat#217004 |
| 15% TBE–urea gels | NOBEX, Invitrogen | Cat#EC6885BOX |
| RNeasy MinElute Cleanup Kit | Qiagen | Cat#74204 |
| QIAquick PCR Purification Kit | Qiagen | Cat#28106 |
| **Experimental models** | | |
| BJ/hTERT | Gift from Anders H. Lund laboratory (Disa Tehler). | N/A |
| HeLa | ATCC | CCL-2 |
| HEK293 | ATCC | CRL-1573 |
| HCT116 | ATCC | CCL-247 |
| MDA-MB-231 | ATCC | HTB-26 |
| **Software** | | |
| BBMap [v38.22] | Bushnell B. | https://sourceforge.net/projects/bbmap/ |
| FastQC [v0.11.4] | Andrews S. | https://www.bioinformatics.babraham.ac.uk/projects/fastqc |
| SAMtools [v1.3.1] | (Li *et al*, 2009) | http://samtools.sourceforge.net |
| tRNAscan-SE [v2.0] | (Chan & Lowe, 2019) | http://lowelab.ucsc.edu/tRNAscan-SE |
| BEDtools [v2.27.1] | (Quinlan & Hall, 2010) | https://bedtools.readthedocs.io/en/latest |
| Segemehl [v0.3.1] | (Hoffmann *et al*, 2009) | https://www.bioinf.uni-leipzig.de/Software/segemehl |
| Picard [v2.18.17] | Broad Institute | https://github.com/broadinstitute/picard |
| GATK [v3.8] | (McKenna *et al*, 2010) | https://software.broadinstitute.org/gatk |
| GSEA [v3.0] | (Subramanian *et al*, 2005) | https://http://software.broadinstitute.org/gsea |
| BLAST [v2.9.0] | (Altschul *et al*, 1990) | https://blast.ncbi.nlm.nih.gov |

## Methods and Protocols

### Cell lines

The cell lines included in this study are HeLa, HEK293, HCT116, MDA-MB-231, and fibroblast BJ/hTERT. The sex of each cell line is as follows: HeLa, female; HEK293, female; MDA-MB-231, female; HCT116, male; and BJ fibroblasts, male. Cells were maintained at 37°C in a humidified atmosphere at 5% $CO_2$ in DMEM 4.5 g/l Glucose with UltraGlutamine media supplemented with 10% of FBS and 1% penicillin/streptomycin.

### RNA extraction

Cells were grown in 60 mm dishes for 48 h. Total RNA from HeLa, HEK293, HCT116, MDA-MB-231, and fibroblast BJ/hTERT was extracted using the miRNeasy Mini kit. Independent replicates were grown, and RNA was extracted on different days. 20 μg of total RNA

was treated following either the protocol of Hydro-tRNAseq (Gogakos *et al*, 2017a) or generic small RNA-seq.

### Hydro-tRNA sequencing

Total RNA was resolved on 15% Novex TBE–urea gels and size-selected for 60-100 nt fragments. The recovered material was then alkaline hydrolyzed (10 mM sodium carbonate and 10 mM sodium bicarbonate) for 10 min at 60°C. The resulting RNA was de-phosphorylated with Antarctic Phosphatase (New England Biolabs) at 37°C for 1 h. De-phosphorylated RNA was purified with an RNeasy MinElute spin column and re-phosphorylated with polynucleotide kinase (NEB). PNK-treated tRNAs were purified with an RNeasy MinElute spin column and, similar to small RNA-seq library preparation, adaptor-ligated, reverse-transcribed, and PCR-amplified for 14 cycles. The resulting cDNA was purified using a QIAQuick PCR Purification Kit and sequenced on Illumina HiSeq 2500 platform in 50 bp paired-end format. Hydro-tRNAseq data of HCT116, MDA-MB-231, and fibroblast BJ/hTERT have been generated in this study, while sequencing data from HEK293 and HeLa had been previously published (Data ref: Benisty *et al*, 2019b).

From all five cell lines, the isoacceptor abundances of MDA-MB-231 yielded a median of 3–5 times higher standard deviation than the other Hydro-tRNAseq quantifications (Table EV2), thus suggesting some technical problem with this cell line. In consequence, this cell line was excluded from any further analysis.

### Small RNA sequencing

Total RNA was directly adaptor-ligated, reverse-transcribed, and PCR-amplified for 12 cycles. The resulting cDNA was then size-selected by gel electrophoresis, and fragments of 145–160 bp were eluted and sequenced on Illumina HiSeq 2500 platform in 50 bp single-end format.

### The Cancer Genome Atlas multi-omics data

Raw small RNA sequencing data in BAM format were retrieved from the GDC legacy archive after obtaining the necessary permissions from dbGaP, comprising all healthy samples (NT, solid tissue normal) and their primary tumor (PT) counterparts, which consists of 23 cancer types (BRCA, PRAD, KICH, KIRP, KIRC, LUAD, LUSC, HNSC, UCEC, CESC, LIHC, CHOL, THCA, COAD, READ, ESCA, STAD, BLCA, PAAD, THYM, SKCM, PCPG, and GBM). For samples for which more than one BAM was available, all files were downloaded. BAM files were converted to FASTQ using SAMtools [v1.3.1] (Li *et al*, 2009). We retrieved publicly available and pre-processed mRNA-seq gene expression, 450k DNA methylation, bisulfite DNA methylation, and SNP6 segmented copy number alteration (CNA) data from firebrowse. As for proteomics, pre-processed protein assembly data and protein relative abundance were obtained from CPTAC for TCGA samples including BRCA, COAD, and READ.

### tRNA quantification and modification calling

In both Hydro-tRNAseq and small RNA-seq FASTQ files, sequencing adapters were trimmed using BBDuk from the BBMap toolkit [v38.22] (https://sourceforge.net/projects/bbmap): k-mer = 10 (allowing 8 at the end of the read), Hamming distance = 1, length = 10–50 bp, and Phred > 25. Using the human reference genome GRCh38 (Genome Reference Consortium Human Reference 38, GCA_000001405.15), a total of 856 nuclear tRNAs and 21

mitochondrial tRNAs were annotated with tRNAscan-SE [v2.0] (Chan & Lowe, 2019).

Trimmed FASTQ files were then mapped using a specific pipeline for tRNAs (Fig 1A) (Hoffmann *et al*, 2018). Summarizing, an artificial genome is first generated by masking all annotated tRNA genes and adding pre-tRNAs (i.e., tRNA genes with 3′ and 5′ genomic flanking regions) as extra chromosomes. Upon mapping to this artificial genome with Segemehl [v0.3.1] (Hoffmann *et al*, 2009), reads that map to the tRNA-masked chromosomes or to the tRNA flanking regions are filtered out in order to remove non-tRNA reads and unmature-tRNA reads, respectively.

After this first mapping step, a second library is generated by adding 3′ CCA tails and removing introns from tRNA genes. All 100% identical sequences of this so-called *mature* tRNAs are clustered to avoid redundancy. Next, the subset of filtered reads from the first mapping is aligned against the clustered mature tRNAs using Segemehl [v0.3.1] (Hoffmann *et al*, 2009). Mapped reads are then realigned with GATK IndelRealigner [v3.8] (McKenna *et al*, 2010) to reduce the number of mismatching bases across all reads.

For quantification, isoacceptors were quantified as reads per million (RPM). In order to increase the coverage for anticodon-level quantification, we consider all reads that map unambiguously to a certain isoacceptor, even though they ambiguously map to different isodecoders (i.e., tRNA genes that differ in their sequence but share the same anticodon). Ambiguous reads mapping to genes of different isoacceptors were discarded.

Regarding modification site calling, we only considered gene-level uniquely mapped reads, as described to be optimal in Hoffmann *et al* (2018). As in their pipeline, in order to distinguish mapping or sequencing errors from true misincorporation sites, we use GATK UnifiedGenotyper [v3.8] (McKenna *et al*, 2010). Furthermore, given that tRNAs have variable D-loop and V-region, we map the detected modifications to the standard tRNA model to make them comparable. We align our tRNA library to the structurally annotated human tRNAs from tRNAdb (Jühling *et al*, 2009) using BLAST [v2.9.0] (Altschul *et al*, 1990) and fit the secondary structure annotation of the top BLAST hits.

### Translational efficiency analysis

#### Relative codon usage (RCU) and relative anticodon abundance (RAA)

The RCU/RAA is defined as the contribution of a certain codon/anti-codon to the amino acid it belongs to. The RCU of all synonymous codons and the RAA of all anticodons recognizing synonymous codons therefore sum up to 1.

$$RCU = \frac{x_C}{\sum_{i \in C_{aa}} x_i} \qquad RAA = \frac{x_A}{\sum_{i \in A_{aa}} x_i}$$

where $x_C/x_A$ refers to the abundance of the codon/anticodon $C/A$, and $C_{aa}$ is the set of all synonymous codons, as well as $A_{aa}$ is the set of all anticodons that decode synonymous codons.

#### tRNA adaptation index (tAI)

As described by dos Reis *et al* (2003, 2004), the tAI weights every codon based on the wobble-base codon–anticodon interaction rules. Let $c$ be a codon, then the decoding weight is a weighted sum of the square-root-normalized tRNA abundances $tRNA_{cj}$ for all tRNA isoacceptors $j$ that bind with affinity $(1-s_{cj})$ given the wobble-base pairing

rules $n_c$. However, while dos Reis *et al* (2004) assume that highly expressed genes are codon-optimized, here we use the non-optimized s-values to avoid a circularity in our reasoning:

$$s = [0, 0, 0, 0, 0.5, 0.5, 0.75, 0.5, 0.5]$$

$$w_c = \sum_{j=1}^{n_c} (1 - s_{cj}) tRNA_{cj}$$

And therefore, the tAI of a certain protein is the product of weights of each codon $i_k$ at the triplet position $k$ throughout the full gene length $l_g$, and normalized by the length.

$$tAI = \left( \prod_{k=1}^{l_g} w_{i_k} \right)^{1/l_g}$$

For this and all further analyses, the coding sequences of *Homo sapiens* from RefSeq were downloaded from the Codon/Codon Pair Usage Tables (CoCoPUTs) project release as of February 6, 2019 (Alexaki *et al*, 2019; Athey *et al*, 2017).

### Relative tRNA adaptation index (RtAI)

For comparison with the SDA (Fig 3A), an amino-acid-normalized tAI measure is defined by dividing each tAI weight by the maximum weight among all codons within each amino acid family.

$$Rw_c = \frac{w_c}{\max_{i \in c_{aa}}(w_i)}$$

And therefore, the RtAI of a certain protein is the product of weights $Rw$ of each codon $i_k$ at the triplet position $k$ throughout the full gene length $l_g$, and normalized by the length.

$$RtAI = \left( \prod_{k=1}^{l_g} Rw_{i_k} \right)^{1/l_g}$$

### Supply-to-Demand Adaptation (SDA)

The SDA aims to consider not only tRNA abundances, but also the codon usage demand. In doing so, it constitutes a global measure of translation control, since the efficiency of a certain codon depends both on its complementary anticodon abundance and the demand for such anticodon by other transcripts. This global control has been indeed established to play an important role in defining optimal translation programs (Frumkin *et al*, 2018).

The definition of the SDA is based on similar previously published metrics (Gingold *et al*, 2012; Pechmann & Frydman, 2013), which consists of a ratio between the anticodon supply and demand. On the one hand, the anticodon supply is defined as the relative tAI weights $Rw$ (see previous section). On the other, the anticodon demand is estimated from the codon usage at the transcriptome level. It is computed as the frequency of each codon in a transcript weighted by the corresponding transcript expression and finally summing up over all transcripts. Let $c$ be a codon, then the codon usage is a weighted sum of the counts of codon $c_i$ in gene $j$ weighted by the mRNA-seq abundance $mRNA_j$ for all genes in the genome $g$:

$$CU_c = \sum_{j=1}^{g} c_{ij} mRNA_j$$

Similarly to the supply, the anticodon demand is then normalized within each amino acid family:

$$D_c = \frac{CU_c}{\max_{i \in c_{aa}}(CU_i)}$$

Finally, the SDA weights (SDAw) are defined as the ratio between the codon supply $S_c$ and demand $D_c$:

$$SDAw_c = \frac{S_c}{D_c}$$

And therefore, the SDA of a certain protein is the product of weights $SDAw$ of each codon $i_k$ at the triplet position $k$ throughout the full gene length $l_g$, and normalized by the length.

$$SDA = \left( \prod_{k=1}^{l_g} SDAw_{i_k} \right)^{1/l_g}$$

### Gene set enrichment analysis (GSEA)

Gene sets derived from the GO Biological Process Ontology were downloaded from the Molecular Signatures Database [v6.2] (MSigDB) as a GMT file (Liberzon *et al*, 2015; Subramanian *et al*, 2005). We analyzed the enrichment of gene sets using the GSEA algorithm (Subramanian *et al*, 2005). The score used to generate the ranked list input is specified in the text for each analysis.

### Survival analysis

To analyze how the supply-to-demand ratio of a certain codon (SDAw) can affect the survival probability in cancer, patients of a certain cancer type are divided in two groups of low/high SDAw, which correspond to the patients having the top and bottom 40% SDAw. The Kaplan–Meier curves are then computed to estimate the survival probability of each group along time.

### tRNA methylation and copy number

For consistency with the current version of publicly available and pre-processed 450k DNA methylation and SNP6 segmented CNA data from firebrowse, we used the human reference genome GRCh37/hg19 (Genome Reference Consortium Human Reference 37, GCA_000001405.1) in this analysis. The coordinates of all nuclear tRNA genes were obtained using tRNAscan-SE [v2.0] (Chan & Lowe, 2019).

Regarding DNA methylation, we computed the average beta value of each tRNA gene from 1.5 kb upstream of the transcription start site (1500TSS) until the end of the gene. For CNA, we retrieved the segmented data of precomputed $log_2(CN)–1$ from firebrowse and extracted the corresponding value for the genomic coordinates containing the tRNA genes. Whenever the tRNA genes were located between two segments, the weighted average in function of the gene overlap with each segment was computed.

### Bisulfite sequencing methylation

As 1500TSS methylation of tRNA genes leads to an average coverage of only 37% genes, we also analyzed the recently published bisulfite sequencing data of 47 samples across nine cancer types (Table EV4; Zhou *et al*, 2018). After retrieving the datasets from the GDC legacy archive, given the higher resolution of bisulfite sequencing data, we restricted the computation of the average promoter methylation of tRNA genes to the GRCh37/hg19 genomic coordinates containing the tRNA genes, since the promoter region of Pol-III-genes is intragenic.

### Multiple linear regression (MLR)

We fitted a multiple linear regression (MLR) between the square-root-normalized tRNA abundance (dependent variable) and the promoter methylation and gene copy number (independent variables). To make all three layers of information comparable, we considered only samples for which all data were available and performed the regression at the isoacceptor level, thus averaging the methylation and CNA data over all tRNA genes that shared the same anticodon.

$$exp = \beta_0 + \beta_{Me}Me + \beta_{CNA}CNA$$

We fitted the model parameters for all 64 isoacceptors and 22 cancer types, leading to $22 \times 64 = 1{,}408$ MLRs, among which only significant coefficients (FDR-corrected *t*-statistic *P*-value < 0.05) were considered in downstream analyses.

### Statistical analysis

For hypothesis testing, an unpaired two-tailed Wilcoxon rank-sum test was performed, unless stated otherwise. All details of the statistical analyses can be found in the Results section. We used a significance value of 0.05. In differential expression analyses, a false discovery rate correction was used to account for multiple testing.

## Data availability

The datasets and computer code produced in this study are available in the following databases:

- Scripts for analyzing tRNA data of TCGA: GitHub (https://github.com/hexavier/tRNA_TCGA).
- Scripts for tRNA mapping: GitHub (https://github.com/hexavier/tRNA_mapping).
- Generated TCGA data (tRNA abundances, SDA, CNA, and DNA methylation): Synapse syn20640275 (www.synapse.org/tRNA_TCGA).
- Hydro-tRNA and small RNA sequencing data of all five cell lines: Gene Expression Omnibus GSE137834 (https://www.ncbi.nlm.nih.gov/geo/query/acc.cgi?acc = GSE137834).

**Expanded View** for this article is available online.

### Acknowledgements

We thank Eva Maria Novoa Pardo, Samuel Miravet-Verde, and Marc Weber for stimulating and critical discussions. We thank the CRG Genomics Unit for assistance with RNA sequencing services. The results published here are in part based on data generated by the TCGA Research Network: https://www.cancer.gov/tcga. We acknowledge the support of the Spanish Ministry of Science and Innovation (MICINN), "Centro de Excelencia Severo Ochoa", the CERCA Programme/Generalitat de Catalunya, and the Spanish Ministry of Science and Innovation (MICINN) to the EMBL partnership. The work of X.H. has been supported by a PhD fellowship from the Fundación Ramón Areces.

### Author contributions

Conceptualization, XH-A, MHS, and LS; Methodology, XH-A, HB, MHS, and LS; Software, XH-A; Investigation, HB and XH-A; Validation, XH-A and MHS; Formal analysis, XH-A and MHS; Writing—Original Draft, XH-A; Writing—Review and Editing, XH-A, HB, MHS, and LS; Visualization, XH-A and MHS; Funding Acquisition, LS; and Supervision, MHS and LS.

### Conflict of interest

The authors declare that they have no conflict of interest.

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
