## [Review Process File · Molecular Systems Biology]

Translational efficiency across healthy and tumor tissues is proliferation-related

Xavier Hernandez-Alias, Hannah Benisty, Martin Schaefer and Luis Serrano.

Review timeline:	Submission date:	2 nd October 2019
	Editorial Decision:	19 th November 2019
	Revision received:	13 th January 2020
	Editorial Decision:	6 th February 2020
	Revision received:	11 th February 2020
	Accepted:	11 th February 2020

Editor: Maria Polychronidou

Transaction Report:

1st Editorial Decision

19th November 2019

Thank you again for submitting your work to Molecular Systems Biology. We have now heard back from the three referees who agreed to evaluate your study. As you will see below, the reviewers acknowledge that the presented findings seem interesting. They raise however a series of concerns, which we would ask you to address in a major revision.

Without repeating all the points listed below, some of the more fundamental issues are the following:

- Reviewer #2 mentions that the effect of tRNA modifications and tRNA fragments on using small RNAs as a proxy for tRNAs needs to be examined further.
- Reviewer #3 mentions that further validations and better statistical support needs to be provided. They provide constructive suggestions in this regard.

All other issues raised by the reviewers would need to be convincingly addressed. Please feel free to contact me in case you would like to discuss in further detail any of the issues raised.

REFeree REPORTS

Reviewer #1:

In this paper in which Serrano and colleagues analyze the tRNA pool in diverse human tissues. They develop new measure of translation efficiency to show that across healthy and tumor tissues it is governed by proliferation. They show that aberrant translation efficiency of two codons ProCCA and GlyGGT in cancer, among other codons is associated with poor patient survival.

I learned a lot and enjoyed very much reading this paper. The analyses of state of the art, performed with rigor and accuracy. The findings are novel and exciting. They provide the strongest indication thus far from the role of proliferation state of tissues in governing their tRNA pool and translation efficiency. The implications onto cancer, in terms of prognosis and in terms of basic biology of translation are novel and illuminating too. Methods are executed with accuracy and suitability. I do not think the paper is very well written, my comments are minor as I'd mainly require clarifications along the following lines:

1. Figure 1B reports Spearman correlation. I think they should report side by side the Pearson correlations as well
2. The report on tRNA modification is a little odd (Fig. 1C) - all we are told is the number of modifications in each sample, nothing about their nature - which modifications are detected, what's the method of detection and quantification of a modification. This is a missed opportunity to convey important information to the reader.
3. The authors report "significant tissue-specificity for all 46 annotated anticodons" - but it's unclear what is the computation method to quantify such "tissue specificity", and how p-value is computed.
4. I don't understand the drawing and method of color coding of Fig 2A...
5. Please define exactly the matrix upon which the PCA is Fig 2B is done.
6. I'm not entirely sure how the "proliferative state of a tissue" is defined, computed or measured.
7. In Fig. 3A I think Pearson correlation will be more appropriate. In fact I'd also appreciate scatter plots that show how well protein quantification and protein to mRNA correlate with tAI, RTE and RtAI
8. The promoter methylation analysis is intriguing, but please clarify - is this CpG methylation or methylation of histones? (I guess the former, but the later was show to play a role too).

Reviewer #2:

This work uses the TCGA data to analyze the correlation of tRNA-mRNA expression (anticodon-codon), addressing an important question in the translation field. The authors first try to establish the concept that small RNA-seq results may be used as proxies for tRNA expression. This conceptual establishment is necessary because only small RNA-seq data, but no tRNA-seq data is available in TCGA. The idea of using small RNA-seq result to infer tRNA expression has been previously reported (Zhang et al., 2018, 2019). The author's new addition to this idea is to directly compare small RNA-seq with hydro-tRNA-seq results from the same cell lines which nicely go beyond the published work. The analysis of the TCGA data identified proliferation as a primary driver of tRNA-mRNA expression, and two specific codons that have large effects on survival.

Major comments:

1. The cell type dependent distribution and functional importance of tRNA fragments are only mentioned in passing, and completely ignored. tRNA fragments are part of the small RNA group and naturally produced for multiple known functions independent of translation. This omission makes the use of small RNA result as proxy of tRNA expression difficult to evaluate. For example, it is known that full tRNA tend to have higher levels of modifications. Would separating the small RNA result according to the extent of modifications (at least for those detectable by sequencing) makes a significant difference in using small RNA-seq as proxy of tRNA expression?
2. Fig. 1 and p. 17 small RNA sequencing protocol: I do not understand how the reported procedure would produce a substantial amount of small RNA reads. Typically small RNA-seq first goes through a size selection step to enrich true biological small RNAs which apparently was not done here.
3. Fig. 1B: the meaning of "significance" is unclear. This crucial data has a y-axis range from 0.90-0.96, which could be interpreted as no difference.
4. There is no raw seq-data presentation (e.g. IGV-like graphs) for comparing small RNA-seq and hydro-tRNA-seq results, making it difficult to evaluate the quality of the seq-data.

Minor comments:

1. For TCGA data, it is misleading to always refer the analysis as tRNA expression. At best the authors used a proxy parameter to infer tRNA abundance. This needs to be made clear throughout.
2. p.12, top: ProCCA is more likely decoded by tRNA^{TGG}, not by tRNA^{AAG}. More explanation is needed on why the authors focus on the analysis of tRNA^{AAG}.

Reviewer #3:

This manuscript quantifies tRNA from small RNA sequencing data, collected without specific measures to capture tRNAs, through the TCGA consortium. These small RNA samples provide a large compendium of small RNA sequencing data, taken from healthy tissues as well as cancer cells. The tRNA levels quantified in different tissues vary substantially, and one dimension of cell-to-cell variation in tRNA abundance correlated with relative expression of Ki67, a marker of proliferation, and more generally with proliferation-related genes. These tRNA levels are used to compute the match between tRNA supply and demand, the latter determined by codon usage in the abundance-weighted transcriptome. Profiles of these supply-versus-demand ratios also cluster by tissue, and correlate with proliferation. This same analysis is applied to tumor samples, where specific codons are seen to shift in the tumor relative to healthy tissue. Within tumors, the "tumor-like" tRNA bias is associated with worse prognosis than the "healthy-like" tRNA bias. Increased tRNA abundance correlates with increased tRNA copy number and reduced promoter methylation.

The impact of changing tRNA pools on cell physiology is an important open question, and our understanding is limited by the lack of large-scale tRNA abundance measurements. Measurement of tRNA abundance from small RNA sequencing data offers a promising solution to this problem, already explored to some extent in the Zhang et al. manuscripts. One key innovation in the present manuscript is an improved mapping approach for tRNAs, which presents a substantial technical challenge. The importance of this innovation for accurate tRNA measurements and its ability to resolve cell-type-specific tRNA differences is not fully validated, however, and I also have concerns about some of the statistical analyses carried out using these measurements. These concerns limit the potential impact of this work.

1. The comparison between the mapping approach proposed here and in Zhang et al. 2018 is presented as simple numerical correlation coefficients in a table. It seems valuable to present these data as actual scatter plots that would demonstrate the dynamic range and other properties of these measurements.
2. It is argued from Figure 1B that the method here captures between cell type tRNA abundance differences. However, in many cases, the distributions of "mismatch vs match" correlations encompass a range that can be substantially higher than the "match" correlation. This seems to raise serious concerns about the ability to capture cell-type-specific tRNA profiles.
3. The manuscript defines a measure of codon demand-to-supply and describes it as a "relative translation efficiency". The term "translation efficiency" has another, specific meaning in this field its use here to describe this tRNA ratio may confuse readers into thinking that it reflects something about ribosome occupancy on the mRNA -- or even that this demand-vs-supply ratio drives this difference. At best, there is a very modest correlation (Figure 2A) with very weak predictive power from which no causal inference may be drawn.
4. This "RTEw" measurement is used substantially through the paper, but it is never tested whether "RTEw" (which includes sample-specific tRNA demand) outperforms simple tRNA abundance.
5. The correlation between promoter methylation and tRNA abundance is interpreted to reflect a causal link, but these could both arise as distinct consequences of some other, underlying cause that reduces tRNA expression and permits or promotes methylation too.

POINT-BY-POINT RESPONSE TO REFEREES**Reviewer #1:**

In this paper in which Serrano and colleagues analyze the tRNA pool in diverse human tissues. They develop new measures of translation efficiency to show that across healthy and tumor tissues it is governed by proliferation. They show that aberrant translation efficiency of two codons ProCCA and GlyGGT in cancer, among other codons is associated with poor patient survival.

I learned a lot and enjoyed very much reading this paper. The analyses of state of the art, performed with rigor and accuracy. The findings are novel and exciting. They provide the strongest indication thus far from the role of proliferation state of tissues in governing their tRNA pool and translation efficiency. The implications onto cancer, in terms of prognosis and in terms of basic biology of translation are novel and illuminating too. Methods are executed with accuracy and suitability.

We thank the reviewer for appreciating the novelty and potential implications of the manuscript findings, as well as the appropriateness of the methods.

I do not think the paper is very well written, my comments are minor as I'd mainly require clarifications along the following lines:

- 1. Figure 1B reports Spearman correlation. I think they should report side by side the Pearson correlations as well*

Although we agree that Pearson correlation is commonly used to establish linear relations between variables, it is nevertheless poorly suited for non-normal distributions with outliers. In Fig R1, we show that tRNA abundances have indeed a long-tail distribution, since few tRNA account for most of the reads while most tRNA are lowly expressed. To circumvent this problem, we tried applying different transformations (logarithmic, square-root), but the resulting correlations are highly transformation-dependent and still biased towards a few highly expressed isoacceptors. For this reason, we used Spearman correlation, which does not assume any specific underlying distribution of the data, in the main text of the original version of the manuscript and prefer to not add a comparison with Pearson correlation in the revised version. Instead,

and in concordance with Fig 1B, we further show cell-line differences with a correlation-independent method. First, in order to assess the amount of tRNA variability coming from either the sequencing method or the cell lines, a Principal Component Analysis of these tRNA quantifications indicates that both factors influence variability to a similar extent (Fig EV1B, >30% variance each). Furthermore, to validate that small RNA-seq is similarly informative of cell type differences as Hydro-tRNAseq, we show that a Linear Discriminant Analysis of the data is able to discriminate between cell lines regardless of the sequencing protocol (Fig EV1B). We hope that this additional analysis will eliminate any doubts regarding the use of small RNA-seq data in detecting cell-line-specific tRNA profiles.

Figure R1. Scatter plot of the absolute tRNA quantifications of HEK293 from small RNA-seq and Hydro-tRNAseq data. The average Reads Per Million (RPM) among the triplicates are plotted, as well as the histogram distribution of all isoacceptors along each axis.

2. *The report on tRNA modification is a little odd (Fig. 1C) - all we are told is the number of modifications in each sample, nothing about their nature - which modifications are detected, what's the method of detection and quantification of a modification. This is a missed opportunity to convey important information to the reader.*

We agree with the reviewer that a deeper look at the nature and patterns of tRNA modifications will emphasize the message of concordance between small RNA-seq and Hydro-tRNAseq data. Modifications had already been determined by modification site calling, as described in the Methods section. We therefore mapped our tRNA library to the human-annotated consensus tRNA structure described in tRNAdb (Jühling *et al*, 2009). From there, the modification patterns of all cell lines and for both sequencing methods were determined position-by-position (Table EV3). In the case of HEK293, the modification pattern of small RNA-seq and Hydro-tRNAseq, as well as their concordances, have been depicted on an illustrative tRNA model in Fig EV3. As now specified in the main text [1st Results section, 3rd paragraph], although the exact nature of modifications cannot be directly determined from sequencing, the most abundant mismatch patterns are in agreement with known tRNA modifications such as adenosine-to-inosine editing and 1-methylinosine. As compared in Table EV3, most known mismatch-producing modifications could be

retrieved by both small RNA-seq and Hydro-tRNAseq. We hope these additional figure and dataset provide the reader with sufficient information regarding the nature and quantification of tRNA modifications, as well as the limitations of the method.

3. *The authors report "significant tissue-specificity for all 46 annotated anticodons" - but it's unclear what is the computation method to quantify such "tissue specificity", and how p-value is computed.*

We have rewritten the sentence to clarify the method and outcomes of this analysis [2nd Results section, 1st paragraph]. Now it reads: "For all 46 annotated anticodons, tRNA abundances have significant differences between tissues, as detected by Kruskal-Wallis test ($q < 0.05$, FDR-corrected)".

4. *I don't understand the drawing and method of color coding of Fig 2A...*

We apologize for not having included a clearer color legend in this figure (as well as in Fig 2B and Fig 3B). We have updated these figures to make the color code clearer.

5. *Please define exactly the matrix upon which the PCA is Fig 2B is done.*

We have included a clearer reference towards the data that was used in Fig 2B [2nd Results section, 2nd paragraph]. Now it reads (changes in italics): "Regarding codon usage, a measure of tRNA abundance taking into account the relative contribution of each tRNA anticodon among the set of *synonymous* codons of a certain amino acid is the Relative Anticodon Abundance (*see Methods*). *Using this measure*, a principal component analysis (PCA) of *all* healthy control samples in TCGA also shows clear differences between tissues (Fig 2B)".

6. *I'm not entirely sure how the "proliferative state of a tissue" is defined, computed or measured.*

We have slightly rearranged the paragraph on the proliferative state of tissues [2nd Results section, 2nd paragraph] in order to clarify the two approaches we use to measure this association. In the revised version, we now (1) introduce the GSEA pointing to proliferation, and (2) validate this link by correlation with Ki67. In addition, we have also better delimited our claims in the last paragraph of this section (changes in italics): "Furthermore, *based on both the gene set enrichment and the association to a proliferation marker*, our analyses identify the proliferative state of tissues as the major biological function driving the variability on tRNA abundances".

7. *In Fig. 3A I think Pearson correlation will be more appropriate. In fact I'd also appreciate scatter plots that show how well protein quantification and protein to mRNA correlate with tAI, RTE and RtAI*

We think that a Pearson correlation would have been appropriate to compare protein quantification to tAI, RTE and RtAI, as proteome abundance is normally distributed and linearly associated. However, regarding the protein-to-mRNA ratios, given the heavy-tailed distribution of quotients, it results in having few outliers biasing the whole Pearson correlation. In consequence, we decided to take Spearman correlations in the main text of the original article, as it does not assume any specific underlying distribution of the data. However, in order to clearly show the data distribution that justifies the method of choice, while at the same time reporting Pearson correlation in parallel whenever possible, as suggested by the reviewer, we have included two additional figures. On the one hand, Fig EV4A shows the raw scatter plots of one of the breast cancer samples, together with the corresponding linear regression wherever a Pearson correlation is suited. On the other, in Fig EV4B, the analogous Pearson analysis to Fig 3A contains the correlations of tAI, RTE

and RtAI against proteome abundances (but not against protein-to-mRNA ratios, for the reason explained above).

8. *The promoter methylation analysis is intriguing, but please clarify - is this CpG methylation or methylation of histones? (I guess the former, but the later was show to play a role too).*

It is indeed DNA methylation, which typically occurs at CpG dinucleotides. While we agree that it would be interesting to investigate the effect of histone methylation on tRNA expression, this data is not available for TCGA samples. To avoid confusion, we have clarified the type of studied methylation at the beginning of the section (changes in italics: "To determine the underlying mechanisms driving changes in expression, we retrieve the *DNA* methylation (*typically occurring at CpG dinucleotides*) and copy number alteration (CNA) data from TCGA samples, as a possible means for tRNA gene regulation").

Reviewer #2:

This work uses the TCGA data to analyze the correlation of tRNA-mRNA expression (anticodon-codon), addressing an important question in the translation field. The authors first try to establish the concept that small RNA-seq results may be used as proxies for tRNA expression. This conceptual establishment is necessary because only small RNA-seq data, but no tRNA-seq data is available in TCGA. The idea of using small RNA-seq result to infer tRNA expression has been previously reported (Zhang et al., 2018, 2019). The author's new addition to this idea is to directly compare small RNA-seq with hydro-tRNA-seq results from the same cell lines which nicely go beyond the published work. The analysis of the TCGA data identified proliferation as a primary driver of tRNA-mRNA expression, and two specific codons that have large effects on survival.

We thank the reviewer for acknowledging the novelty of the work.

Major comments:

1. *The cell type dependent distribution and functional importance of tRNA fragments are only mentioned in passing, and completely ignored. tRNA fragments are part of the small RNA group and naturally produced for multiple known functions independent of translation. This omission makes the use of small RNA result as proxy of tRNA expression difficult to evaluate. For example, it is known that full tRNA tend to have higher levels of modifications. Would separating the small RNA result according to the extent of modifications (at least for those detectable by sequencing) makes a significant difference in using small RNA-seq as proxy of tRNA expression?*

We are aware of the presence of tRNA-derived fragments (tRFs) in small RNA-seq datasets, and completely agree with the reviewer's concerns. In this context, we have made some changes and performed further analyses to address them:

- ❖ We studied the nature and patterns of detected tRNA modifications along the consensus tRNA structure. Modifications had already been determined by modification site calling, as described in the Methods section. We therefore mapped our tRNA library to the human-annotated consensus tRNA structure described in tRNAdb (Jühling *et al*, 2009). From there, the modification patterns of all cell lines and for both sequencing methods were determined position-by-position (Table

EV3). In the case of HEK293, the modification pattern of small RNA-seq and Hydro-tRNAseq, as well as their concordances, have been depicted on an illustrative tRNA model in Fig EV3.

- ❖ As suggested by the reviewer, we analyzed the correlation of small RNA-seq reads depending on the extent of modified/mismatched residues. Using HEK293 as the model cell line for this analysis, we divided small RNA-seq reads in four categories:
 - All reads (see Methods section): it includes all reads that can unambiguously be assigned to a certain isoacceptor, even though they are ambiguously mapped to several isodecoders.
 - Unique reads: reads that are unambiguously mapped to a certain isodecoder.
 - Matched reads: unique reads that do not contain mismatches.
 - Reads with mismatches: unique reads that contain mismatches.

As shown in Table R1, no significant differences between reads with or without mismatches are found when compared to tRNA-seq protocols in which tRFs are specifically removed before sequencing. In consequence, while we still cannot avoid having tRFs in our datasets, we can indeed prove they are not substantially biasing the quantification.

Table R1. Spearman correlation of tRNA quantifications from small RNA-seq data against conventional tRNA-seq protocols. The tRNA abundances from HEK293 were quantified in Reads Per Million (see Methods) and averaged among the triplicates. Different subsets of the small RNA-seq reads were compared to all other tRNA-seq datasets.

Small RNA-seq (this paper)	Hydro-tRNAseq (this paper)	Hydro-tRNAseq (Gogakos et al , 2017)	Hydro-tRNAseq (Mattijssen et al , 2017)	DM-tRNAseq (Zheng et al , 2015)
All reads	0.94	0.85	0.78	0.80
Unique reads	0.83	0.75	0.69	0.78
Matched reads	0.84	0.76	0.70	0.79
Reads with mismatches	0.83	0.72	0.65	0.75

- ❖ We now explicitly mention tRFs in the main text [1st Results section, 1st paragraph]: "It has been reported that there are tRNA-derived fragments naturally produced and having other functions different from translation (Schimmel, 2018), which could confound the tRNA quantification. Although we cannot exclude the presence of tRNA-derived fragments in small RNA-seq datasets (Torres et al, 2019), we found that no differences between reads with or without mismatches are found when compared to tRNA-seq protocols in which tRFs are specifically removed before sequencing".

Overall, all our analyses indicate that (1) there is no clear modification pattern related to small RNA-seq compared to mature tRNAs, and (2) small RNA-seq quantifications highly correlate with tRNA-seq protocols. We hope we could address the main concerns regarding tRFs, and appreciate the reviewer comment in order to clarify this issue in the main text.

2. *Fig. 1 and p. 17 small RNA sequencing protocol: I do not understand how the reported procedure would produce a substantial amount of small RNA reads. Typically small RNA-seq first goes through a size selection step to enrich true biological small RNAs which apparently was not done here.*

We thank the reviewer for bringing this point up, and apologize for the mistake in the original manuscript. There is indeed a size-selection step to enrich for small RNAs. We have corrected the Methods section, so it now reads (changes in italics): "Total RNA was directly adaptor-ligated, reverse-transcribed and

PCR-amplified for 12 cycles. *The resulting cDNA was then size-selected by gel electrophoresis, and fragments of 145-160 bp were eluted and sequenced on Illumina HiSeq 2500 platform in 50bp single-end format*".

3. *Fig. 1B: the meaning of "significance" is unclear. This crucial data has a y-axis range from 0.90-0.96, which could be interpreted as no difference.*

We agree that all correlations, even for mismatching pairs, are very high. However, in this paper, we want to emphasize that among the distribution of human tRNA abundances, different tissues and cell states showcase different profiles even if they differ just in a few isoacceptors. From this perspective, and in concordance with the reviewer, such small tRNA changes can often be diluted in a global correlation analysis as in Fig 1B. To better address this, we performed additional analyses to elucidate the amount of tRNA variability coming from either the sequencing method or the cell lines. First, a Principal Component Analysis of these tRNA quantifications indicates that both factors influence variability to a similar extent (Fig EV1B, >30% variance each). Furthermore, to validate that small RNA-seq is similarly informative of cell type differences as Hydro-tRNAseq, we show that a Linear Discriminant Analysis of the data is able to discriminate between cell lines regardless of the sequencing protocol (Fig EV1B). We hope that this additional evidence on cell-type-specific profiles will clarify the extent of tRNA differences.

4. *There is no raw seq-data presentation (e.g. IGV-like graphs) for comparing small RNA-seq and hydro-tRNA-seq results, making it difficult to evaluate the quality of the seq-data.*

In order to report the quality of our sequencing data (both small RNA-seq and Hydro-tRNAseq), we have included read length and coverage statistics in Fig EV2. In Fig EV2A, we plot the read length distribution in all cell lines in their triplicates. In Fig EV2B, we show, for each isoacceptor, the fraction of tRNA sequence that is covered by 1 or more reads. Finally, Fig EV2C contains the IGV-like graph of tRNAPro^{AGG}, as one of the most studied tRNAs along the manuscript. We hope this supplementary figure can now guide the reader in evaluating and comparing the performance of our sequencing datasets.

Minor comments:

1. *For TCGA data, it is misleading to always refer the analysis as tRNA expression. At best the authors used a proxy parameter to infer tRNA abundance. This needs to be made clear throughout.*

As suggested by the reviewer, we tried to emphasize that all TCGA quantifications were determined from small RNA-seq data. In this context, we modified the main text [2nd Results section, 1st paragraph] as follows (changes in italics): "To determine the tissue-specificity of tRNAs in physiological conditions, the tRNA levels of all 675 healthy samples in TCGA tissues are analyzed *from small RNA-seq data*". In addition, we removed all references to our small RNA-seq quantifications as "tRNA expression" throughout the whole text, using "tRNA abundance", "tRNA quantification", "tRNA level", etc., instead. Finally, we also extended the Discussion to explicitly raise this point of debate [1st paragraph]: "All in all, even though our quantifications from small RNA-seq just give an estimate of the tRNA abundances, the results indicate that they can be rather precise proxies".

2. *p.12, top: ProCCA is more likely decoded by tRNATGG, not by tRNAAGG. More explanation is needed on why the authors focus on the analysis of tRNAAGG.*

We agree with the reviewer that the analysis of tRNAPro^{AGG} instead of tRNAPro^{TGG} in recognizing ProCCA deserved a better clarification. We have extended the Results section [5th Results section, 3rd paragraph] to explicitly discuss this rationale (changes in italics): "Given the association of the codon ProCCA with cancer prognosis (Fig 4C), we explore the *abundance* pattern of tRNAPro in TCGA. *While*

both tRNA^{Pro}^{TGG} and tRNA^{Pro}^{AGG} are able to decode ProCCA, the latter specifically appears overexpressed in 8 out of 9 cancer types (Fig EV5A), making it a candidate driver of the translational differences".

Reviewer #3:

This manuscript quantifies tRNA from small RNA sequencing data, collected without specific measures to capture tRNAs, through the TCGA consortium. These small RNA samples provide a large compendium of small RNA sequencing data, taken from healthy tissues as well as cancer cells. The tRNA levels quantified in different tissues vary substantially, and one dimension of cell-to-cell variation in tRNA abundance correlated with relative expression of Ki67, a marker of proliferation, and more generally with proliferation-related genes. These tRNA levels are used to compute the match between tRNA supply and demand, the latter determined by codon usage in the abundance-weighted transcriptome. Profiles of these supply-versus-demand ratios also cluster by tissue, and correlate with proliferation. This same analysis is applied to tumor samples, where specific codons are seen to shift in the tumor relative to healthy tissue. Within tumors, the "tumor-like" tRNA bias is associated with worse prognosis than the "healthy-like" tRNA bias. Increased tRNA abundance correlates with increased tRNA copy number and reduced promoter methylation.

The impact of changing tRNA pools on cell physiology is an important open question, and our understanding is limited by the lack of large-scale tRNA abundance measurements. Measurement of tRNA abundance from small RNA sequencing data offers a promising solution to this problem, already explored to some extent in the Zhang et al. manuscripts. One key innovation in the present manuscript is an improved mapping approach for tRNAs, which presents a substantial technical challenge. The importance of this innovation for accurate tRNA measurements and its ability to resolve cell-type-specific tRNA differences is not fully validated, however, and I also have concerns about some of the statistical analyses carried out using these measurements. These concerns limit the potential impact of this work.

We thank the reviewer's comments in acknowledging the research niche of the manuscript and its approach. We are happy to address all concerns in order to strengthen the message and impact of the article.

- 1. The comparison between the mapping approach proposed here and in Zhang et al. 2018 is presented as simple numerical correlation coefficients in a table. It seems valuable to present these data as actual scatter plots that would demonstrate the dynamic range and other properties of these measurements.*

We did not include a deeper comparison between mapping approaches in the original submission, as the absolute tRNA quantifications from Zhang et al (2018) of HEK293 were not publicly available. Upon data request to the authors, we have now included a more accurate and complete correlation table (Table EV1). Furthermore, as suggested by the reviewer, we include the raw scatter plots corresponding to these correlations (Fig EV1A).

- 2. It is argued from Figure 1B that the method here captures between cell type tRNA abundance differences. However, in many cases, the distributions of "mismatch vs match" correlations encompass a range that can be substantially higher than the "match" correlation. This seems to raise serious concerns about the ability to capture cell-type-specific tRNA profiles.*

We agree that the correlation range of "mismatching" distributions is often large, and even sometimes overlapping with "matching" pairs. This is a consequence of the different sizes of each group, since "matching" distributions contain n=9 points while "mismatching" ones have n=63 points. Therefore, while the Wilcoxon rank-sum test is well suited for unequal sample sizes, it becomes also clear that, among all "mismatching" pairs, not all cell lines are equally distinct or similar to each other.

Furthermore, all correlations, even for mismatching pairs, are very high. In this context, we want to emphasize that among the distribution of human tRNA abundances, different tissues and cell states showcase different profiles even if they differ just in a few isoacceptors. From this perspective, and in concordance with the reviewer, such small tRNA changes can often be diluted in a global correlation analysis as in Fig 1B.

To tackle both issues raised above, we performed additional analyses to elucidate the amount of tRNA variability coming from either the sequencing method or the cell lines. First, a Principal Component Analysis of these tRNA quantifications indicates that both factors influence variability to a similar extent (Fig EV1B, >30% variance each). Furthermore, to validate that small RNA-seq is similarly informative of cell type differences as Hydro-tRNAseq, we show that a Linear Discriminant Analysis of the data is able to discriminate between cell lines regardless of the sequencing protocol (Fig EV1B). We hope that this additional evidence on cell-type-specific profiles will clarify the extent of tRNA differences.

- 3. The manuscript defines a measure of codon demand-to-supply and describes it as a "relative translation efficiency". The term "translation efficiency" has another, specific meaning in this field its use here to describe this tRNA ratio may confuse readers into thinking that it reflects something about ribosome occupancy on the mRNA -- or even that this demand-vs-supply ratio drives this difference. At best, there is a very modest correlation (Figure 2A) with very weak predictive power from which no causal inference may be drawn.*

We agree with the reviewer in that any effect of the supply-to-demand ratio in determining the process of translation is at most modest, as stated in the Results and Discussion sections. In order to emphasize this more clearly, we chose another name along the line of other terms alike (tAI, CAI). All references to the previous "RTE" have been modified to "Supply-to-Demand Adaptation" (SDA).

Furthermore, to clarify any mention to "translational efficiency", we have defined it the first time it appears [3rd Results section, 1st paragraph]: "The so-called translational efficiency is defined as the rate of protein production from mRNA, and multiple indices and models can be described to estimate it (Gingold & Pilpel, 2011). In this article . . .". With this introduction, we hope to make it clear to the reader that SDA is at most an estimator for translational efficiency, and not confounded with actual ribosome occupancy.

- 4. This "RTEw" measurement is used substantially through the paper, but it is never tested whether "RTEw" (which includes sample-specific tRNA demand) outperforms simple tRNA abundance.*

Before going into the core of the question, we would like to make a distinction between the concepts of "tRNA abundance" and "RTE" (now called SDA): while tRNA abundance refers to the levels of the 46 known tRNA isoacceptors, the SDA is a measure of adaptation of all 61 amino-acid-encoding codons (partially based on their cognate tRNA abundances). In this context, and following the line of discussion of the previous answer, we propose the SDA as an alternative to other metrics of tRNA-based codon adaptation, such as the tRNA Adaptation Index (tAI). As such, we believe the results in Fig 3A show how SDA performs compared to other measures alike. To highlight this more clearly, we have now added to the

main text [3rd Results section, 2nd paragraph]: "In consequence, including the mRNA codon demand into the SDA metric outperforms other tRNA-only metrics of translational efficiency".

5. *The correlation between promoter methylation and tRNA abundance is interpreted to reflect a causal link, but these could both arise as distinct consequences of some other, underlying cause that reduces tRNA expression and permits or promotes methylation too.*

We are aware that such a causal link cannot be established from the correlations and have therefore modified the Results section (changes in italics) [5th Results section, 3rd paragraph]: "Despite the limited explained variance of the models (average $R^2=0.023$), such results *suggest* that promoter methylation *could contribute* to inhibition of tRNA gene expression, whereas an increase in the gene copy number *would enhance* tRNA expression". Similarly, we also revised the Discussion section (changes in italics): "In an effort to elucidate the mechanisms regulating the expression of tRNAs, we observe that the tRNA gene copy number and their DNA methylation state have a positive and inhibitory *association with tRNA abundances*, respectively. . . . Here we specifically propose a role for DNA methylation in regulating the overexpression of tRNA^{Pro}^{AGG} in cancer, *although no direct causal link can yet be established*".

Thank you again for sending us your revised study. We have now heard back from the two referees who were asked to evaluate your study. As you will see below, the reviewers acknowledge that the performed revisions have addressed most of the previously raised issues. However, reviewer #3 still lists a couple of remaining concerns, which can be addressed by text modifications, and we would ask you to address in a minor revision.

REFEREE REPORTS

Reviewer #2:

The authors have adequately addressed my comments.

Reviewer #3:

Revisions to the manuscript have addressed my specific concerns with the original version.

Broadly, I do think the extraction of tRNA reads from size-selected small RNA sequencing introduces the possibility of substantial artifacts. I had not appreciated that this was part of the small RNA sequencing workflow based on the original manuscript. We know that tRNA cleavage produces fragments that fall closer to the small RNA size range and could thus affect abundance in these sequencing data. Further, we know that tRNA cleavage is regulated and so differences in tRNA cleavage could masquerade as differences in tRNA abundance. The validation data presented in the manuscript don't address this point, and it would be very hard to do so comprehensively -- at a minimum, however this potential confounding effect needs to be discussed clearly.

As a minor point, I would suggest revising the abstract to read: "the codon-anticodon co-adaptation in humans is not completely understood, NOR IS its effect on tissue-specific protein levels."

POINT-BY-POINT RESPONSE TO REFEREES**Reviewer #3:**

Revisions to the manuscript have addressed my specific concerns with the original version.

Broadly, I do think the extraction of tRNA reads from size-selected small RNA sequencing introduces the possibility of substantial artifacts. I had not appreciated that this was part of the small RNA sequencing workflow based on the original manuscript. We know that tRNA cleavage produces fragments that fall closer to the small RNA size range and could thus affect abundance in these sequencing data. Further, we know that tRNA cleavage is regulated and so differences in tRNA cleavage could masquerade as differences in tRNA abundance. The validation data presented in the manuscript don't address this point, and it would be very hard to do so comprehensively -- at a minimum, however this potential confounding effect needs to be discussed clearly.

We agree with the reviewer that this is an important point of discussion of the current article, and we do not intend to ignore the presence of tRNA cleavage in our dataset, nor its importance. In fact, aware of this potential artifact, we already tried to convey the existence of tRNA-derived fragments in the Results section of the previous version (1st section, 1st and 2nd paragraph) as well as in Figure EV1B.

In addition, to bring this point up again in the Discussion, we now explicitly mention the potential interference of tRNA-derived fragments [Discussion, 1st paragraph] (changes in italics): "As a result, we obtain reproducible and informative quantifications of all isoacceptors in our gold standard cell lines as well as in thousands of samples across 23 cancer types of TCGA, exceeding the quality of similarly published data (Zhang et al, 2018, 2019). *However, we cannot exclude that tRNA-derived fragments (tRFs) could be interfering with our small RNA-seq quantifications*".

As a minor point, I would suggest revising the abstract to read: "the codon-anticodon co-adaptation in humans is not completely understood, NOR IS its effect on tissue-specific protein levels."

We have incorporated the suggestion, and appreciate that it indeed clarifies the meaning of the sentence.

Accepted

11th February 2020

Thank you again for sending us your revised manuscript. We are now satisfied with the modifications made and I am pleased to inform you that your paper has been accepted for publication.

Corresponding Author Name: Luis Serrano and Martin Schaefer

Manuscript Number: MSB-19-9275